# Effect of Selected Factors Influencing Biogenic Amines Degradation by *Bacillus subtilis* Isolated from Food

**DOI:** 10.3390/microorganisms11041091

**Published:** 2023-04-21

**Authors:** Irena Butor, Petra Jančová, Khatantuul Purevdorj, Lucie Klementová, Maciej Kluz, Ivana Huňová, Hana Pištěková, František Buňka, Leona Buňková

**Affiliations:** 1Department of Environmental Protection Engineering, Faculty of Technology, Tomas Bata University in Zlín, Nad Ovčírnou 3685, 760 01 Zlín, Czech Republic; irena.burdikova@gmail.com (I.B.); purevdorj@utb.cz (K.P.); klementova@utb.cz (L.K.); i_hunova@utb.cz (I.H.); pistekova@utb.cz (H.P.); bunkova@utb.cz (L.B.); 2Department of Bioenergy, Food Technology and Microbiology, Institute of Food Technology and Nutrition, University of Rzeszow, 4 Zelwerowicza St, 35601 Rzeszow, Poland; mkluz@ur.edu.pl; 3Food Quality and Safety Research Laboratory, Department of Logistics, Faculty of Military Leadership, University of Defence, Kounicova 65, 662 10 Brno, Czech Republic; frantisek.bunka@gmail.com

**Keywords:** biogenic amines, *Bacillus subtilis*, cultivation, degradation

## Abstract

Modern food technology research has researched possible approaches to reducing the concentration of biogenic amines in food and thereby enhance and guarantee food safety. Applying adjunct cultures that can metabolise biogenic amines is a potential approach to reach the latter mentioned goal. Therefore, this study aims to study the crucial factors that could determine the decrease in biogenic amines concentration (histamine, tyramine, phenylethylamine, putrescine and cadaverine) in foodstuffs using *Bacillus subtilis* DEPE IB1 isolated from gouda-type cheese. The combined effects of cultivation temperature (8 °C, 23 °C and 30 °C) and the initial pH of the medium (5.0, 6.0, 7.0 and 8.0) under aerobic and also anaerobic conditions resulted in the decrease of the tested biogenic amines concentration during the cultivation time (another factor tested). *Bacillus subtilis* was cultivated (in vitro) in a medium supplemented with biogenic amines, and their degradation was detected using the high-performance liquid chromatography equipped with UV-detector. The course of biogenic amines degradation by *Bacillus subtilis* DEPE IB1 was significantly influenced by cultivation temperature and also the initial pH of the medium (*p* < 0.05). At the end of the cultivation, the concentration of all of the monitored biogenic amines was significantly reduced by 65–85% (*p* < 0.05). Therefore, this strain could be used for preventive purposes and contributes to food safety enhance.

## 1. Introduction

Biogenic amines (BA) are organic nitrogen bases of low molecular weight that occur naturally in living organisms as metabolic intermediates and products. The compounds are synthesised and degraded during metabolism in animals, plants, and microorganisms. BA are mainly formed by the decarboxylation of amino acids or by the amination and transamination of aldehydes and ketones [1,2]. The chemical structure of BA can be aliphatic (putrescine, cadaverine, spermine, spermidine), aromatic (tyramine, phenylethylamine) or heterocyclic (histamine, tryptamine) [3]. They play an important role in many human physiological functions, such as cerebral activity, gastric acid secretion and immune responses [4].

BA may accumulate in food in high concentrations via activities of microorganisms that possess decarboxylation enzymes. Excessive oral intake of BA can cause nausea, headache, rashes, and changes in blood pressure [4]. The BA content in food fluctuates as many factors contribute to their formation both during the production process and when food is stored. One of the most important factors is the quality and condition of the raw input material, and in the case of fermented products, a suitable selection of added starter cultures [5,6]. Foods that often contain elevated levels of BA are fish, fish products and derivatives and fermented products [7,8,9]. Especially exogenous histamine represents one of the triggers of scombroid food poisoning [10]. Cheese is one of the most common fermented foods associated with higher level of BA [11,12], and the term ‘cheese reaction’ is used for tyramine intoxication, when concentrations are generally higher than 1 g·kg^−1^ of the body weight. Tyramine and histamine were found to be the most common types of BA in meat and dairy products [13,14], putrescine was the most common in a variety of fruits, juices, and vegetables [15].

Due to adverse effects on health, BA accumulation in foods should be prevented [16], and it is important to develop new and effective strategies to eliminate these substances. Several approaches have been suggested, such as inhibiting bacteria that produce BA, reducing the number of BA producers using heat treatment of raw milk for cheese production, high-pressure treatment, ionising radiation, packaging technology, using food additives, reducing proteolytic activity (reducing the availability of amino acids—precursors of BA) and controlling food microflora applied, i.e., selecting starter cultures without the significant BA production. Starter cultures are pure or mixed beneficial microorganisms used in fermented food products. Applying starter cultures with amine oxidase (AO) activity is important as it inhibits the formation of BA [17,18]. It has been observed that a rapid decrease in pH using appropriate starter cultures (possessed AO activity and also without the significant intrinsic decarboxylase activity) can largely prevent the accumulation of BA in fermented foods. Using the latter mentioned starters, such are lactic acid bacteria including lactobacilli and/or cocci could prevent the formation of BA in various fermented products. Starter cultures capable of nutritionally competing with non-starter microorganisms and/or contaminants, especially during the maturation process and storage, can further prevent the excessive production of BA [19]. These methods can prevent, slow, or reduce the formation of BA to an acceptable level but cannot eliminate existing amines. Another method of reducing BA in food is to directly metabolize them using microorganisms or enzymes, which is the only possible method of degrading already formed BA [20]. Using microorganisms to reduce BA is based on the presence of AO, which is responsible for detoxifying the BA taken in food [17,20,21]. In general, AOs are a class of oxidative deaminases that can decompose BA into aldehydes and ammonia. These compounds are widespread in animals, plants and microorganisms and play a crucial role in the process of oxidative deamination and maintenance of body homeostasis in living organisms [22].

It is a generally accepted fact that the ability of microorganisms to degrade BA is strain-specific [23], and the use of appropriate strains appears to be a suitable modern strategy to reduce BA levels in food. On the other hand, it is very difficult to simply prevent the accumulation of BA applying only starters (including lactic acid bacteria that are part of the food’s usual microflora) with insignificant BA-production activities resulting in low BA concentrations in the final stages of the production and/or storage process [24,25]. Up to the moment, he direct eliminates of the already existing BA from food has not been sufficiently studied yet, and the literature on using microorganisms to degrade BA in food is very limited. It is known that some representatives of the of the family *Lactobacillaceae*, gram-positive cocci (e.g., *Pediococcus*, *Micrococcus*, *Staphylococcus*) and other gram-positive rods (e.g., *Bacillus*, *Brevibacterium*) can degrade biogenic amines [26,27] and have been applied to food for a crucial control its BA content [21].

This study aims to investigate an innovative approach to support, enhance and generally maintain food safety using the adjunct culture of *Bacillus subtilis* DEPE IB1 (the BA-metabolizing strain isolated from the gouda-type cheese) and study the (in vitro) degradation processes of the following BA: putrescine, phenylethylamine, histamine, cadaverine and tyramine. Additionally, the effect of selected factors (such as temperature [8 °C, 23 °C and 30 °C], pH-value [pH ∈〈5; 8〉] and aerobic or anaerobic conditions) were tested for supporting of the main goal and identifying the optimal conditions for maximizing of BA elimination. The successfully applied strains able to BA degradation could significantly contribute to enhance of food safety. Generally, *Bacillus subtilis* strains are able to grow in many foodstuffs under relatively wide conditions. Therefore, the incorporate of the latter mentioned strains as possible starters represents practically the ideal solution of high BA contents in foodstuffs.

## 2. Materials and Methods

### 2.1. Strain and Cultivation Conditions

The strain of *Bacillus subtilis* DEPE IB1 used in this study was obtained from the bacteria collection at the Department of Environmental Protection Engineering (DEPE) at Tomas Bata University in Zlín and was isolated from ripened gouda-type cheese. Initially, the strain was routinely cultured at 30 ± 1 °C for 48 h in Nutrient Broth (HiMedia, Mumbai, India). For selected factors tested, the above mentioned strain was cultivated in so called “a liquid mineral medium” (see below) contained also an individually dosed biogenic amine (putrescine, phenylethylamine, histamine, cadaverine and tyramine). Each biogenic amine was tested separately.

### 2.2. Sample Preparation for the Analysis of BA Degradation Depending on Cultivation Conditions

The combined effect of selected factors on the decrease in BA was observed during cultivation in a liquid mineral medium (MM1) containing KH_2_PO_4_ solution (working solution 0.9% (*w*/*v*) in a volume of 20 mL in 1 litre of MM1), Na_2_HPO_4_∙12 H_2_O solution (working solution 2.4% (*w*/*v*) in a volume of 80 mL in 1 litre of MM1), Mg(SO_4_)∙7 H_2_O (10 g·L^−1^), Fe(NH_4_)_2_(SO_4_)_2_∙6 H_2_O (3 g·L^−1^), CaCl_2_∙2 H_2_O (1 g·L^−1^), NaCl (50 g·L^−1^), a micronutrient solution (2 mL·L^−1^) composed of the following elements: MnSO_4_∙5 H_2_O (0.043 g·L^−1^), H_3_BO_3_ (0.057 g·L^−1^), ZnSO_4_∙7 H_2_O (0.043 g·L^−1^), (NH_4_)_6_Mo_7_O_24_∙4 H_2_O (0.037 g·L^−1^), Co(NO_3_)_2_∙6 H_2_O (0.025 g·L^−1^), CuSO_4_∙5 H_2_O (0.040 g·L^−1^) and a solution for BA testing (putrescine, phenylethylamine, histamine, cadaverine and tyramine, separately). The final concentration of each biogenic amines was 2 g·L^−1^. The individual BA were studied separately. All components were purchased from Sigma Aldrich (St. Louis, MI, USA). The medium’s pH was adjusted to 1 mol·L^−1^ HCl or NaOH, and four variants were prepared: pH 5.0 ± 0.1, 6.0 ± 0.1, 7.0 ± 0.1 and 8.0 ± 0.1. In a case of each BA tested, 5 mL of the medium was transferred into 864 tubes (4 pH values ✕ 3 temperatures ✕ 2 environments ✕ 6 sampling times [including the time “zero”] ✕ 6 repeated), which were inoculated with 100 µL of pre-prepared inoculum (24-h culture; (4.6 ± 0.3).10^7^ CFU). Half the samples (432 tubes) were cultivated under aerobic condition; the other were incubated under anaerobic condition. The anaerobic environment was achieved by covering the culture medium with sterile paraffin oil (1 mL; Lach:ner, Neratovice, Czech Republic).

Degradation was monitored at three temperatures (30 ± 1, 23 ± 1 and 8 ± 1 °C), and the samples were cultivated at different sampling times and combinations of factors in six parallel tubes. Different sampling times were determined for each temperature. At 8 °C, samples were taken at 12, 24, 48, 72 and 96 h of cultivation; at 23 °C, samples were taken at 6, 12, 24, 48 and 72 h; at 30 °C, samples were taken at 6, 12, 24, 36 and 48 h. At different incubation temperatures, the sampling times were chosen and performed so that the cells were at approximately the same growth phase. For 5 BA, totally 4320 tubes (5 BA ✕ 864 tubes for each BA) were analysed. After incubation of the tested strain, the medium was centrifuged (3500× *g*; 22 ± 1 °C; 20 min), and the acquired supernatant was diluted (1:1; *v*/*v*) with perchloric acid (1.2 mol·L^−1^; Sigma-Aldrich, St. Louis, MI, USA). The acidified mixture was filtered (0.22 µm).

### 2.3. HPLC Analysis of Biogenic Amines

The degrading capacity of the *Bacillus subtilis* DEPE IB1 strain was tested in a mineral medium by HPLC/UV.

The acidified mixture was subjected to derivatisation according to Dadáková et al. [28], and 1,*7*-diaminoheptane (500 mg·L^−1^; Sigma-Aldrich, St. Louis, MI, USA) was used as an internal standard. The derivatised samples were filtered (0.22 µm) and applied on the column (ZORBAX RRHD Eclipse Plus C18, 50 × 3.0 mm, 1.8 µm, Agilent Technologies, Santa Clara, CA, USA) of a chromatographic system (pump, autosampler, degasser, UV/VIS-DAD detector [λ = 254 nm] and column thermostat [Thermo Fisher Scientific, Waltham, MA, USA]). The conditions for the separation and detection of BA are described by Jančová et al. [29] and successfully applied, e.g., in microbiological studies of Purevdorj et al. [30]. Each of the six broths prepared for one tested microorganism was derivatised twice, and each derivatised mixture was applied on the column twice (n = 24).

### 2.4. Statistical Analysis

The Kruskal–Wallis and Wilcoxon tests were used to evaluate the differences between BA occurrence in individual samples. The statistical program Unistat 6.5 (Unistat Ltd., London, UK) was used to process the data, and the significance level was 0.05.

## 3. Results and Discussion

This study evaluated an innovative approach to eliminate BA using *B. subtilis* to degrade five selected BA under different culture conditions. It examined the influence of temperature, the initial pH of the medium and method of cultivation (under aerobic and anaerobic conditions) for degradation activity over time. A reduction in all monitored BA was observed during the experiment.

The results of degradation activity under anaerobic cultivation are summarised in Figure 1A,C,E,G, Figure 2A,C,E,G, Figure 3A,C,E,G, Figure 4A,C,E,G and Figure 5A,C,E,G. The highest rate of degradation was recorded at higher temperatures (*p* < 0.05). At 23 °C and pH 7, there was a significant decrease (*p* < 0.05) in the levels of all monitored BA 24 h after inoculation, their amounts were reduced by more than 60%.

After 72 h of cultivation, out of all variants of the initial pH of the medium, the greatest decrease was observed in putrescine (Figure 2), but compared to other amines, the resulting concentrations were not significantly lower. A significant decrease in BA was exhibited only 12 h after inoculation at 30 °C in all variants of the medium’s pH level. After 48 h of cultivation at this temperature, histamine, putrescine and cadaverine were reduced by 80%. None of the initial pH mediums tested had substantial inhibitory effects on degradation activity (*p* < 0.05). After anaerobic cultivation for 72 h at 23 °C, at pH 5, the amount of tyramine degraded to 30% (Figure 5A). With a higher pH, slightly higher tyramine levels were detected: 33% at pH 6 and pH 7 and 36% at pH 8. The opposite effect was observed under aerobic cultivation at 8 °C, as putrescine was reduced to 16% at pH 5 over 96 h, while at pH 7, putrescine levels decreased to 11% (Figure 2). Under anaerobic cultivation at 8 °C, there was no effect on the degradation of phenylethylamine, which had degraded to 30% at pH 5 and pH 8.

All of the monitored BA were reduced by at least 59% after 24 h of cultivation at 8 °C, and at this temperature, *B. subtilis* DEPE IB1 showed the highest degradation activity in the medium with pH 7. The results revealed that a significant decrease (*p* < 0.05) was observed in the BA samples cultivated at 8 °C after 24 h, histamine levels decreased by 65% at pH 5 and pH 7 and by 62% at pH 6. Tyramine concentration dropped by 61% at pH 5, 59% at pH 6, 65% at pH 7 and 62% at pH 8. Levels of phenylethylamine decreased by 40% at pH 5 and pH 6, and at pH 7 and pH 8, phenylethylamine concentration decreased by 38% (Figure 1). The amount of putrescine after 24 h of cultivation decreased by 72% at pH 5, 69% at pH 6 and 70% at pH 7 and pH 8). After 24 h of cultivation at 8 °C the concentration of cadaverine dropped by 35% in all of the monitored pHs (Figure 3). At 8 °C, levels of putrescine decreased after 96 h by 15% (Figure 2). In contrast, phenylethylamine degraded least, and after 96 h, it was detected at 30% less than the original amount (Figure 1).

The results of degradation activity under the aerobic cultivation are shown in Figure 1B,D,F,H, Figure 2B,D,F,H, Figure 3B,D,F,H, Figure 4B,D,F,H and Figure 5B,D,F,H. Aerobic cultivation also revealed reduced levels of all BA monitored. The smallest decrease in concentration occurred in phenylethylamine, and after 96 h, it reduced to 33% of its initial amount (Figure 1). At 8 °C, in the pH 7 medium, putrescine was the most degraded BA under aerobic conditions, and after 96 h of cultivation, its concentration reduced to 12% (Figure 2F). Putrescine and histamine were the most degraded at 23 °C as their levels declined by 80% (Figure 2 and Figure 4). At 30 °C and 48 h after inoculation, the highest reduction rates in histamine, putrescine and cadaverine (Figure 2, Figure 3 and Figure 4) was observed, and their levels reduced to 20%. In this experiment, the chilling temperature had no significant effect on the degradation activity of the tested *B. subtilis* DEPE IB1 strain. However, a decrease in the degradation activity of the test strain with increasing ambient temperature can be expected depending on the optimum growth temperature of *B. subtilis*, which ranges from 30–37 °C [31].

One of the objectives of the experiment was to compare the degradation rate of tested BA by *B. subtilis* DEPE IB1 strain under aerobic and anaerobic conditions. Generally, aerobic cultivation resulted in the same decreased quantity of BA as under anaerobic conditions. Aeration had no significant effect on the degradation rate (*p* ≥ 0.05). Based on the results, it can be concluded that the other tested combination of factors (initial pH of the medium and incubation temperature), with the exception of aerobic/anaerobic environment, significantly affect (*p* < 0.05) the degradative activity of the tested strain, *B. subtilis* DEPE IB1. A decrease in baseline BA was observed for these tested factors as it reduced by 65–85%. In this study, positive results of BA degradation using *B. subtilis* DEPE IB1 were observed along with statistically significant interactions between individual factors (cultivation time, temperature, and pH value; *p* < 0.05). The effects of the combination of these factors, i.e., increasing temperature, increasing pH values and longer cultivation time, resulted in a higher rate of BA degradation. It can be deduced from the obtained results that using this BA degrader could potentially reduce the amount of BA in food when methods of prevention do not preclude their formation.

The issues of using isolation of degraders of BA from food was examined by Leuschner et al. [32], who managed to isolate degraders of tyramine and histamine from food identified as *Lactiplantibacillus* (previously *Lactobacillus*) *plantarum*, *Lacticaseibacillus* (previously *Lactobacillus*) *casei*, *Lactiplantibacillus* (previously *Lactobacillus*) *pentosus*, *Pediococcus acidilactici*, *Rhodococcus* sp., *Arthrobacter* sp., *Micrococcus* sp., *Brevibacterium linens* and *Geotrichum candidum*. After the depletion of free nutrients from the substrate, organisms were forced to use carbon and nitrogen bound in BA, which has been described in the study of Alvarez and Moreno-Arribas [21]; Lee et al. [33]; Sun et al. [34]. Zaman et al. [23] isolated bacteria from fish sauces, which was evaluated by the BA’s degradation activity, and the strain of *Bacillus* genus used in this study degraded one or more BA (among them, *B. subtilis*) and their ability to degrade histamine from 27% to 60%, putrescine from 7% to 30% and cadaverine from 22% to 29% over 24 h under in vitro conditions.

The production of BA in decarboxylase-positive microorganisms is strongly affected by ambient temperature: lower temperatures slow down their production and reduce the total amount of produced BA [20]. The effect of temperature on the activity of tyramine oxidase was observed by Leuschner and Hammes [35]. Reduced activity of the enzyme was observed at 5 °C, while at 37 °C and 40 °C was found the highest degradation activity. At 60 °C and 70 °C, tyramine was not degraded due to the denaturation of the tyramine oxidase. Eom et al. [36] isolated several *Bacillus* strains from fermented soy products that significantly degraded histamine and tyramine. Eight histamine-degrading bacteria were isolated from salted fish products by Lee et al. [37]: *Agrobacterium tumefaciens*, *Bacillus cereus*, *Paenibacillus* (previously *Bacillus*) *polymyxa*, *B. licheniformis*, *B. amyloliquefaciens*, *B. subtilis* and *Rummeliibacillus stabekisii*. Lee et al. [37] also observed a 74% histamine degradation in the *B. subtilis* broth within 24 h, and *Paenibacillus polymyxa* exhibited 100% degradation of histamine. *Bacillus* strains that are able to degrade biogenic amines have also been isolated from other foods, such as various types of cheese [38], shuidouchi [39], or from fish sauce [40].

However, the addition of spore-forming bacteria to foods with a potentially higher risk of BA requires further research to focus on their safety [41], potential negative impact or the sensory properties of the food to which they have been applied. The presence of spore-forming bacteria in food is generally considered undesirable because they are characterised by strong enzymatic facilities, such as *Weizmania* (previously *Bacillus*) *coagulans* and *Geobacillus stearothermophilus*, which cause food to spoil (especially canned non-acidic food), and *Bacillus cereus* may be responsible for food poisoning [42]. Ouoba et al. [43] identified *B. subtilis* and *B. pumilus* as potential starter cultures for the production of African fermented beans. As the tested strain reduced the concentration of BA present under different culture conditions and at different pH values, it could potentially be used in foods that require specific technological procedures during production. The use of biopreservative methods to reduce or prevent the BA accumulation in fermented foods has become a hot research trend, and some methods have already been used in practical production [24,44].

## 4. Conclusions

Degradation of five BA (histamine, tyramine, phenylethylamine, putrescine and cadaverine) was demonstrated under in vitro conditions using *B. subtilis* DEPE IB1 isolated from ripened gouda-type cheese. The initial pH of the culture medium and cultivation temperature significantly affected the degradative activity of the isolated *B. subtilis* DEPE IB1 strain, as the content of all BA studies decreased by 65–85%. Biogenic amines were the most reduced at 30 °C. Cultivation in aerobic and anaerobic conditions did not significantly affect the rate of degradation. The obtained results indicate the possibility of using *B. subtilis* DEPE IB1 strain in food production as the adjunct/protective culture, especially for fermented foods. The degradation potential of the above mentioned microorganism can significantly affect food quality and reduce the risk negative effects on human health, and therefore, increase food safety. This study highlights a new emerging method for reducing the concentration of formed BA in food but also suggests other possible directions for further research in this area.

## Figures and Tables

**Figure 1 microorganisms-11-01091-f001:**
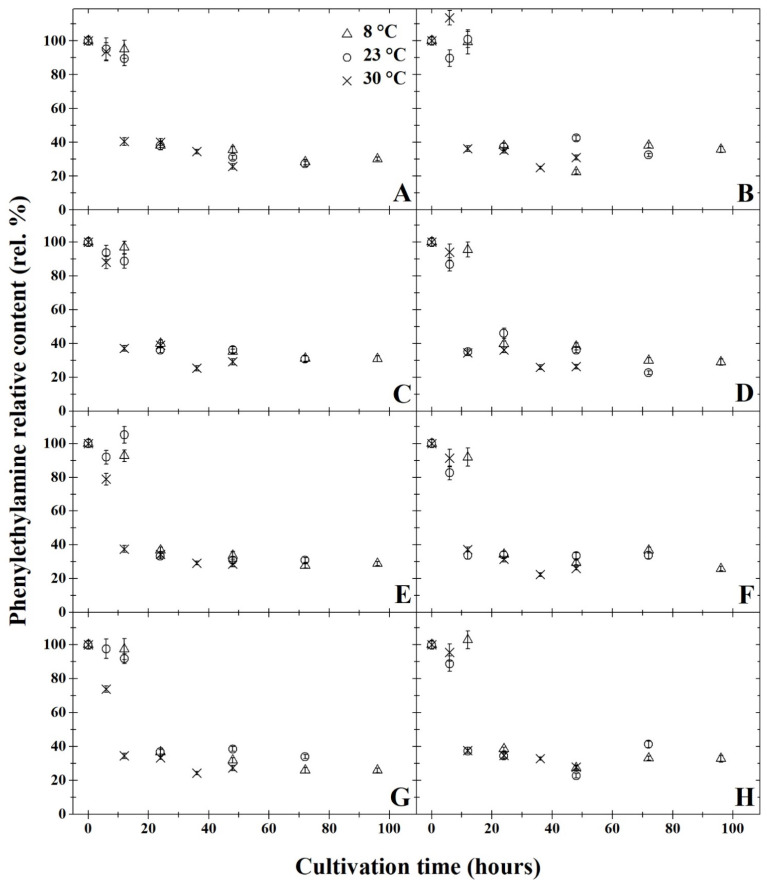
Development of phenylethylamine relative content (rel. %) by degradation using *Bacillus subtilis* DEPE IB1 in dependence on the cultivation time (hours); the initial phenylethylamine relative content at the time zero were always 100%. Parts (**A**,**C**,**E**,**G**) represent the results of anaerobic cultivation and parts (**B**,**D**,**F**,**H**) show the results of aerobic cultivation. Incubation was carried out under pH = 5 (parts **A** and **B**), pH = 6 (parts **C** and **D**), pH = 7 (parts **E** and **F**), pH = 8 (parts **G** and **H**). Three cultivation temperatures were used: 8 °C (△), 23 °C (○) and 30 °C (✕). The results were presented using the means and standard deviations (bars; n = 24).

**Figure 2 microorganisms-11-01091-f002:**
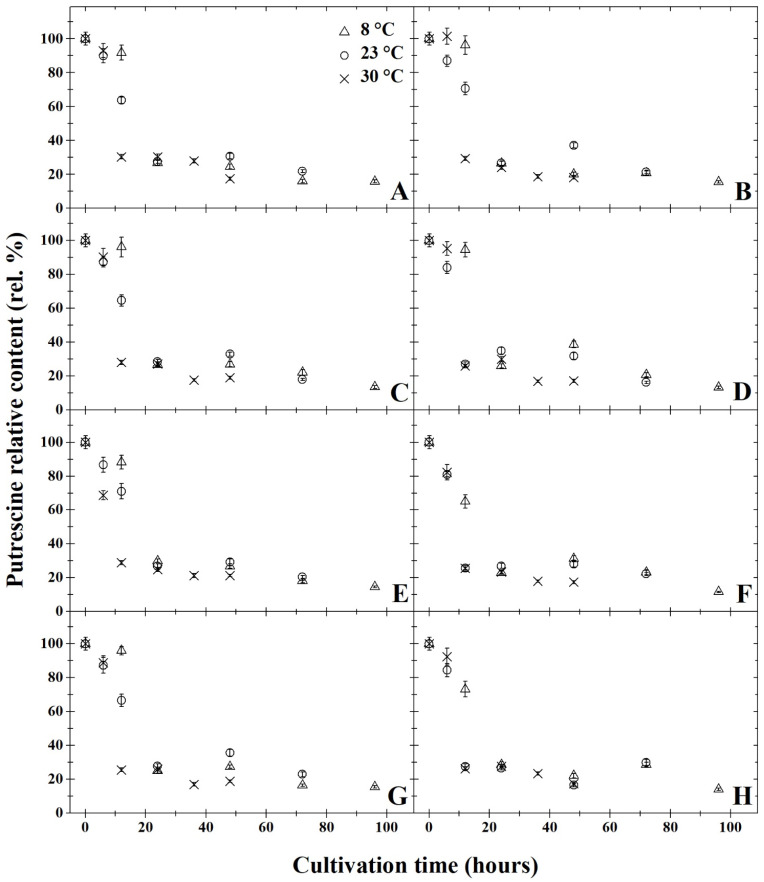
Development of putrescine relative content (rel. %) by degradation using *Bacillus subtilis* DEPE IB1 in dependence on the cultivation time (hours); the initial putrescine relative content at the time zero were always 100%. Parts (**A**,**C**,**E**,**G**) represent the results of anaerobic cultivation and parts (**B**,**D**,**F**,**H**) show the results of aerobic cultivation. Incubation was carried out under pH = 5 (parts **A** and **B**), pH = 6 (parts **C** and **D**), pH = 7 (parts **E** and **F**), pH = 8 (parts **G** and **H**). Three cultivation temperatures were used: 8 °C (△), 23 °C (○) and 30 °C (✕). The results were presented using the means and standard deviations (bars; n = 24).

**Figure 3 microorganisms-11-01091-f003:**
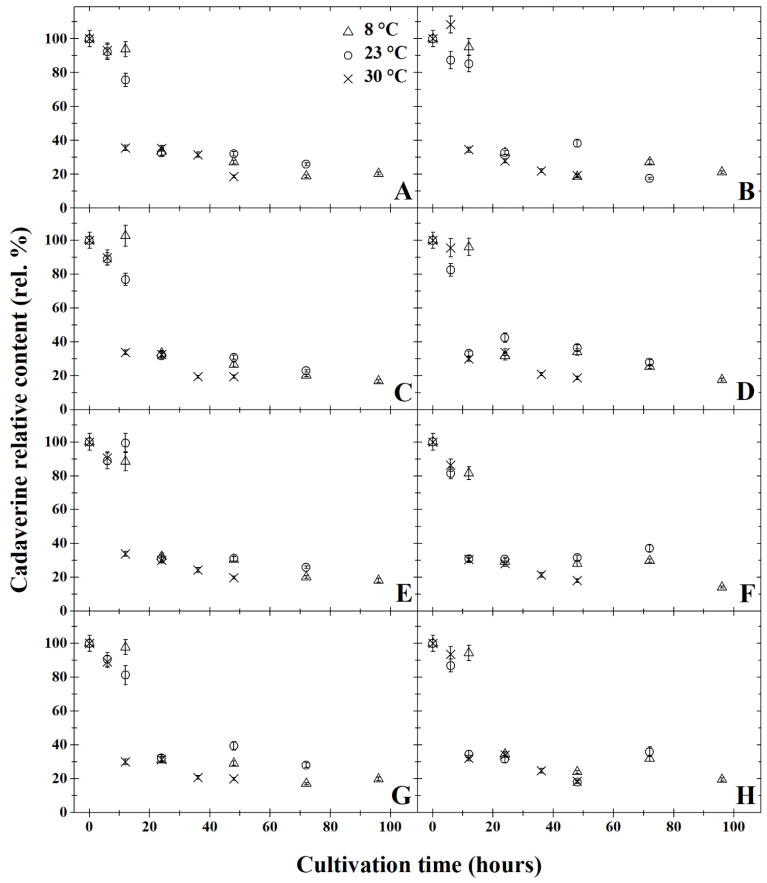
Development of cadaverine relative content (rel. %) by degradation using *Bacillus subtilis* DEPE IB1 in dependence on the cultivation time (hours); the initial cadaverine relative content at the time zero were always 100%. Parts (**A**,**C**,**E**,**G**) represent the results of anaerobic cultivation and parts (**B**,**D**,**F**,**H**) show the results of aerobic cultivation. Incubation was carried out under pH = 5 (parts **A** and **B**), pH = 6 (parts **C** and **D**), pH = 7 (parts **E** and **F**), pH = 8 (parts **G** and **H**). Three cultivation temperatures were used: 8 °C (△), 23 °C (○) and 30 °C (✕). The results were presented using the means and standard deviations (bars; n = 24).

**Figure 4 microorganisms-11-01091-f004:**
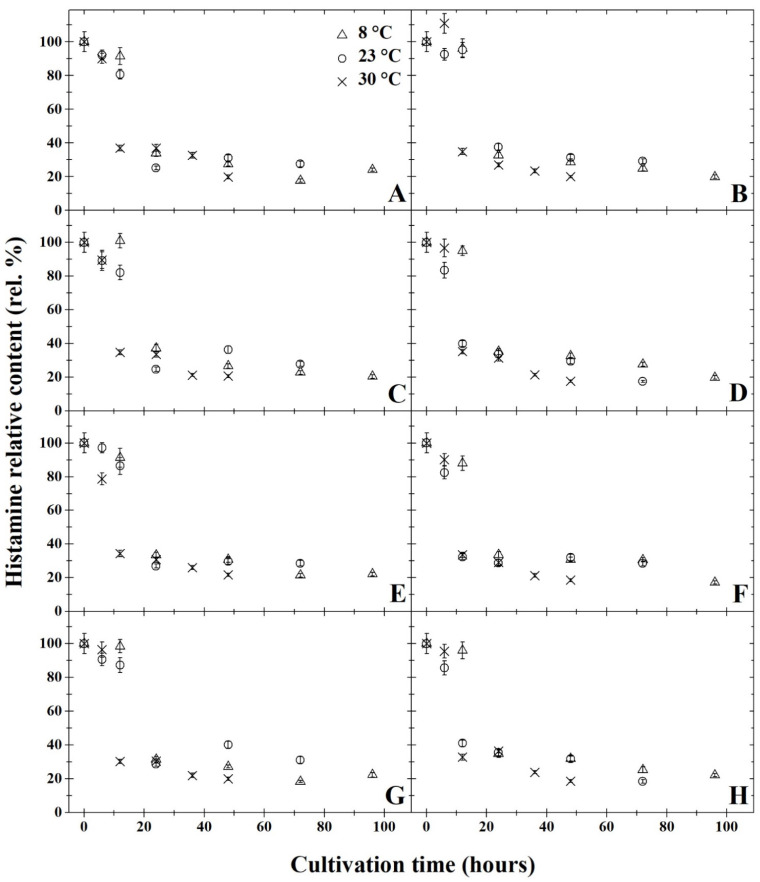
Development of histamine relative content (rel. %) by degradation using *Bacillus subtilis* DEPE IB1 in dependence on the cultivation time (hours); the initial histamine relative content at the time zero were always 100%. Parts (**A**,**C**,**E**,**G**) represent the results of anaerobic cultivation and parts (**B**,**D**,**F**,**H**) show the results of aerobic cultivation. Incubation was carried out under pH = 5 (parts **A** and **B**), pH = 6 (parts **C** and **D**), pH = 7 (parts **E** and **F**), pH = 8 (parts **G** and **H**). Three cultivation temperatures were used: 8 °C (△), 23 °C (○) and 30 °C (✕). The results were presented using the means and standard deviations (bars; n = 24).

**Figure 5 microorganisms-11-01091-f005:**
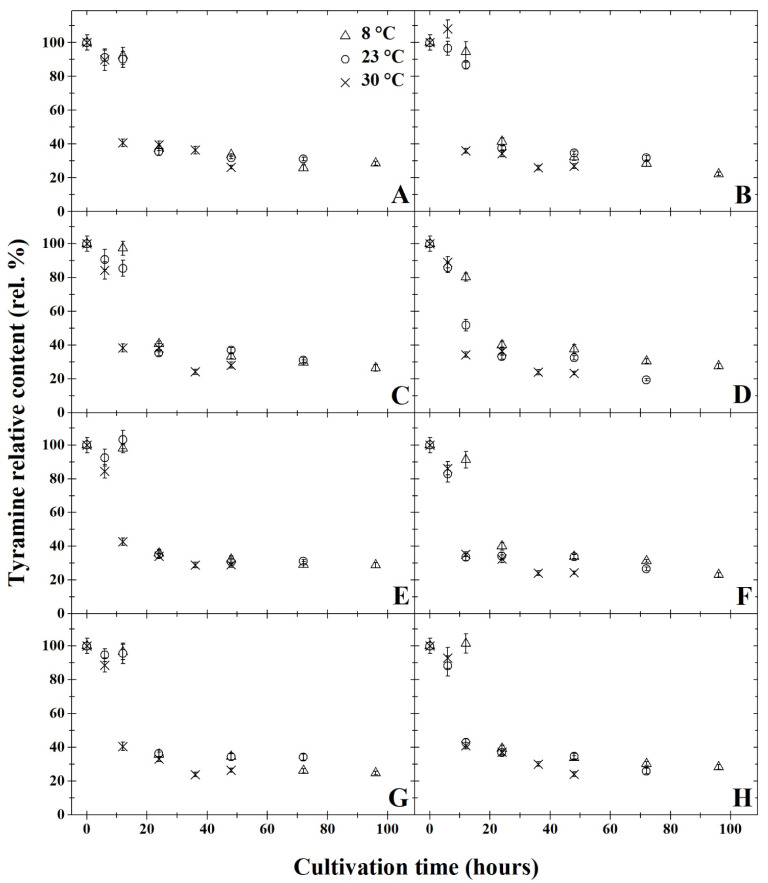
Development of tyramine relative content (rel. %) by degradation using *Bacillus subtilis* DEPE IB1 in dependence on the cultivation time (hours); the initial tyramine relative content at the time zero were always 100%. Parts (**A**,**C**,**E**,**G**) represent the results of anaerobic cultivation and parts (**B**,**D**,**F**,**H**) show the results of aerobic cultivation. Incubation was carried out under pH = 5 (parts **A** and **B**), pH = 6 (parts **C** and **D**), pH = 7 (parts **E** and **F**), pH = 8 (parts **G** and **H**). Three cultivation temperatures were used: 8 °C (△), 23 °C (○) and 30 °C (✕). The results were presented using the means and standard deviations (bars; n = 24).

## Data Availability

Data is contained within the article.

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
