# Peer review of "Effect of Selected Factors Influencing Biogenic Amines Degradation by *Bacillus subtilis* Isolated from Food"

_microorganisms, 2023, doi:10.3390/microorganisms11041091_

Round 1
Reviewer 1 Report
This paper entitled “Effect of selected factors influencing biogenic amines degradation by Bacillus subtilis isolated from food” evaluated the decrease in biogenic amines concentration in foodstuffs using Bacillus subtilis DEPE IB1 isolated from gouda-type cheese. The authors present some interesting and useful information for readers. However, they must clarify some unclear points and some additional information is needed for clarification. Moreover, the manuscript is written in poor English, which needs to be improved, if possible, to check with native English speaker to improve English and Grammar.
Main Comments:
1. L.16, Delete "of" before "the crucial factors...".
2. L.41: " Biogenic amines " revised to "BA".
3. L. 66: "decartoxylase activity" revised to "decarboxylase activity".
4. L.84: "appling " revised to "applying".
5. L.90-91: " biogenic amines " revised to "BA"
6. L.91: "crutial " revised to "crucial".
7. L.98: "conditons" revised to "conditions ".
8. L.99: "conditons" revised to "conditions ".
9. L.130: The sentence "Half the samples (360) were cultivated…". Please check the tubes (864/2=432, not 360?).
10. L.165-166: The sentence is unclear, there is no mention of pH 8 data in histamine levels.
11. L.167-168: The sentence is unclear, please rephrase it.
12. L. 278-279: The sentence started at "At 5oC,..." is unclear, please rephrase it.
13. L.308: "reducet" revised to " reduced".
14. L.412: "Bacillus subtilis" must be italicized.
15. L. 415: " Parkia biglobosa " must be italicized.
Reviewer 2 Report
Comments and Suggestions for Authors (Manuscript ID: microorganisms-2311268)
The aim of the work is veryfication of the use of the Bacillus subtilis DEPE IB1 strain for biogenic amines (BA) degradation and thus improve food safety. The experiment was well planed. The Introduction is well written and provides information about previous research related with presence of BA in food and their detoxification. The aim of the paper has been clearly formulated. Minor revision for the English language is requested.
Some minor suggestions:
-Line 132: lan:cher???
-Lines 135-139: Why were samples not taken for analysis at the same time?/ Different sampling times were determined for each temperature. At 8 °C, samples were taken at 12, 24, 48, 72 and 96 hours of cultivation; at 23 °C, samples were taken at 6, 12, 24, 48 and 72 hours; at 30 °C, samples were taken at 6, 12, 24, 36 and 48 hours./
-Line 158-159 This sentence needs improvement.
- Analyzing the results requires a lot of involvement from the reader. I wonder if the results could not be presented in a different way than using the graphs attached to the work.
-The presented data show that the temperature was significant for BA degradation only in the first hours of cultivation. I would like to ask why the values presented in the graphs (Fig1B; Fig 3BC, 4B, 5B,H) are higher than 100% (basic BA level)?
-The literature used in the work is correctly selected, but would require updating (about 50% are works from before 2011)
References require editorial correction e.g. Line 417 – capital letters must be used; L.334 – dot instead of comma; Line 337: no dot, should be Sci., Line 342: something is missing here?
Round 2
Reviewer 1 Report
The authors have responded properly to my earlier review comments.